# Shooting Prediction Based on Vision Sensors and Trajectory Learning

**Yuliang Zhao [1,2], Xinyue Zhang [1], Mingliang Yang [3], Qingchao Zhang [4], Jian Li [1] , Chao Lian [1,*], Changbo Bi [1], Zhiping Wang [1] and Guanglie Zhang [5,*]**

1 Sensor and Big Data Laboratory, Northeastern University, Qinhuangdao 066000, China
2 Laboratory of Micro-Nano Precision Optical Sensing and Measurement Technology, Qinhuangdao 066000, China
3 Zhejiang Institute of Communication, Hangzhou 311112, China
4 State Key Laboratory of Dynamic Measurement Technology, School of Instrumentation and Electronics, North University of China, Taiyuan 038507, China
5 Shenzhen Research Institute, City University of Hong Kong, Shenzhen 518057, China
* Correspondence: lianchao@neuq.edu.cn (C.L.); gl.zhang@cityu.edu.hk (G.Z.)

**Abstract:** Basketball has become one of the most popular sports and is generally popular in international sports events. However, how to effectively achieve shooting prediction and then guide shooting has become a major challenge. Different from the classical manual observation method, this paper proposes a real-time shooting prediction method based on vision sensors and trajectory learning. In the research, we first extracted the basketball trajectory information on template matching and centroid calculation and then obtained a smooth trajectory curve through interpolation. Taking the change of x, y coordinate position, height Y, and distance D from the shooting point during the instantaneous movement as basic features, four machine learning algorithms were used to analyze the impact of different feature combinations on the shooting prediction. Finally, we analyzed the minimum trajectory point requirements predicted when making a shot. The experimental results show that our method can effectively predict the effect of shooting when the feature combination is basketball height and time. When the interpolation density is high (the total number of trajectory points is 116), the overall accuracy can reach more than 90%, and only one-third of the effective trajectory length is required, which effectively helps athletes improve their shooting percentage and assist referees in daily training.

**Keywords:** image processing; basketball; motion trajectories; training models; shooting prediction

## 1. Introduction

The development of artificial intelligence technology has led to changes in real life and promoted progress in various fields, and the sports industry is no exception [1]. Artificial intelligence technology is powerfully changing sports, and has greatly developed in medical rehabilitation, physical therapy, conditioning training, exercise guidance, etc. [2–6]. Basketball is a popular sport, and the research and application of artificial intelligence and ML in play guidance and sports analysis is very extensive. Artificial intelligence technology allows basketball players to get immediate and long-term feedback in training and let them know how perfectly their movements are being executed. Furthermore, it is also used by coaches to provide reinforcement feedback, which is one of the most powerful factors influencing the motor learning and ability of basketball players [7].

In basketball, shooting is arguably the most important technique, and shooting pattern is the basic indicator for evaluating player performance in professional basketball research. A good shooter can achieve precision shooting from distance, position within the standard area, or available power to avoid blocking shots from close range, which is why shooting technique has been the subject of much research over the years. For example, F. Yin [8]

selected nine players and analyzed the shooting patterns of basketball players in the form of visual charts. To obtain better metrics of basketball performance, recent research efforts have turned to machine learning of wearables data using AI [9,10].

The current research mostly stays on the identification and analysis of the shooting action, and there are few types of research on the shooting percentage prediction. From the point of view of the key equipment used, the research on the recognition and analysis of shooting movements has formed two common ways, namely wearable sensors and video. There are few studies based on wearables. For example, N. Kuhlman [11] collects shooting accelerometer data, analyzes its characteristics, and uses machine learning to build a shooting motion analysis system, which can provide a reference for athletes' shooting motions. It shows that four different shot types can be classified with an accuracy of 86.3% using a Quadratic SVM classifier. However, this approach requires more experimentation and more complex characterization. C. Lian [12] used the smart wristband embedded with sensors to analyze 18 shooting movements and identify the shooting movements. S. Shankar [13] developed a smart device using an IMU and WIFI module, which can record and recognize the shooting action. The device described in the paper can be adapted to an individual's shooting movements and provide accurate real-time improvements during training. However, this method can only judge free throws standing on the free-throw line, and it lacks consideration of other positions. In recent years, shooting action through the analysis of shooting videos has become a hot issue in the field of sports analysis, which can provide assistance for standardizing athletes' movements and improving the effectiveness of basic training. Obtaining the basketball trajectory and analyzing the motion characteristics of the trajectory is the basis for predicting the shooting result. In addition, other researchers have used image processing technology and artificial intelligence technology to analyze players' shooting movements [14,15]. R. Ji proposed a basketball shooting gesture recognition method based on image feature extraction and machine learning. The author mainly uses the method of image feature extraction to collect the action pose data of basketball players and extracts multi-dimensional action pose features of the time domain and frequency domain. After that, the authors achieved accurate classification and recognition of basketball shooting gestures. For upper limb recognition of basketball gestures, the recognition accuracy of the Bayesian algorithm can reach 91.2%. For lower limb recognition of basketball poses, the recognition accuracy of random forest algorithm reaches 94.1%.

Regarding the prediction of shots, the most instructive method is Wang's method [16]. He used the frame difference method to process the shooting video images, and obtained the basketball trajectory from the shooting start point to the shooting end point. After that, he used basic mechanics theory to analyze and calculate the relationship between shooting angle and shooting speed, which provided an effective trajectory prediction method and reference for obtaining shooting percentage. The authors highlight some of the factors that have an impact on shot success. These factors include shot height, shot speed, shot angle, and potential wind speed that may be present. Through experiments, this paper illustrates the effects of different wind directions, wind speeds, shooting angles, shooting distances, and heights on the shooting percentage. Cheng C [17] designed an automatic shooting robot. He improved the shooting percentage through guiding the robot to adjust the shooting angle by means of a laser pointer. In the laboratory, the robot's shooting accuracy reached almost 90%. However, performance may depend on the lighting conditions in the shooting environment. Ref. [18] analyzed the shooting data of NBA players and made shot predictions using two methods including random forest and XGBoost. The shooting prediction rate can reach 68%. Whether it is an emotional factor, a slight loss of balance, or just a small movement out of place, it can affect the outcome of a shot.

Although there have been more and more research works related to basketball players' shooting performance in recent years, there are very few studies on trajectory prediction using the information and data reflected from the set of shooting trajectories. The study found that it is possible to predict information about shooting through the early perfor-

mance information of basketball trajectory changes. This paper attempts to use these data to analyze and predict basketball percentages.

The object of our study is free-throw shooting. Therefore, we specify that the player needs to stand at the free-throw line to make the shot. In this way, we ensure that the player's position with respect to the basket is fixed, and the player faces towards the basket. The shooting trajectory can be considered to be in a two-dimensional plane since there is little motion in the third dimension. Therefore, we can determine a two-dimensional plane from the player to the basket. The features are extracted through the two-dimensional plane. A reasonable shot trajectory will increase the probability of the basketball going into the basket and a greater chance of scoring. Especially the farther away from the basket, the more the athlete should consider the trajectory of the shot. The accurate movement trajectory can help players adjust their shooting posture and strength to improve their free-throw shooting performance. Based on the trajectory learning prediction model, this paper provides a novel and effective method for the instantaneous prediction of shooting results. Collecting a large number of shooting videos, the template matching method and the regional centroid method are used to obtain the motion trajectory coordinates of the basketball in the video. At the same time, a variety of different trajectory motion feature combinations are used to construct feature combinations, and the instantaneous prediction model is trained by the trajectory learning method. Finally, according to the early basketball movement characteristic data after the basketball is shot, the shooting is judged. The contributions of this paper are as follows:

1. A new method using trajectory learning is proposed; we use the early trajectory motion data of basketball trajectories to achieve shot prediction;
2. A high-precision basketball trajectory acquisition method is proposed, which uses instantaneous movement time t, x, y position changes during movement, height Y, and distance from the shooting point D as the basic elements to achieve trajectory prediction;
3. A method to achieve shot prediction through machine learning is proposed, which mainly explores the impact of different feature combinations and different trajectory lengths on shot prediction.

This paper divides the work into six sections for a detailed description: Section 1 mainly introduces the research background and significance of the paper. We summarize some methods for the recognition and analysis of basketball movements through sensors and video, and analyze and study the methods of shooting prediction. Section 2 presents the scheme for capturing shooting videos. Section 3 introduces the methods of obtaining basketball performance trajectories and shooting predictions. Section 4 utilizes different machine learning methods to classify shooting results for different feature combinations in the captured videos. Section 5 introduces the principles of different shot recognition methods and illustrates the limitations of the method proposed in this paper. Section 6 is the conclusions of this paper.

## 2. Shooting Video Collection Scheme

We chose an outdoor basketball venue and invited players with a certain basketball foundation to shoot. In order to improve the accuracy of the experiment, the basketball players were replaced many times and the shooting angle was adjusted, and each video shot was required to contain a complete shooting process, as shown in Figure 1. The camera was a GoPro HERO 6 Black motion camera (produced from Beijing, China), which supports Android and IOS systems, and the video resolution was $1920 \times 1080$, that is, the size of each frame in the video was $1920 \times 1080$. The video frame rate was 120 frames per second, which can meet the definition requirements. In addition, it makes full use of the anti-shake camera performance of GoPro HERO 6 Black, adds slow-motion recording at 1080 p 240 fps, 2.7 K 120 fps, and supports 4K60 and 1080 p 240 dynamic video. The captured video was temporarily stored in the SD card that comes with the camera, ready to be copied to the computer for further processing.

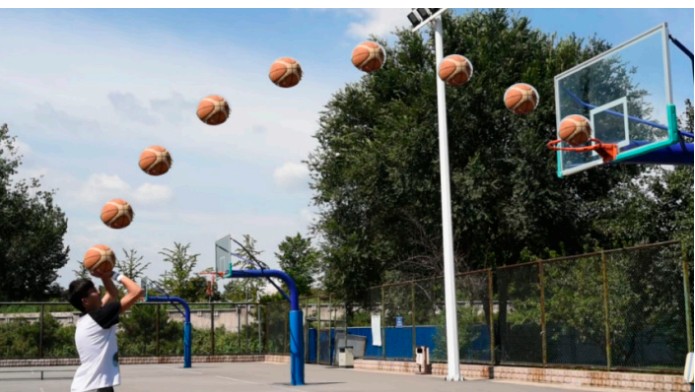

**Figure 1.** The movement of basketball when shooting.

### 3. Basketball Movement Trajectory Acquisition and Shooting Prediction Method

*3.1. Obtaining the Trajectory of the Basketball When Shooting*

The shooting trajectory data is the data basis for constructing the shooting prediction model, and the coordinate position of the basketball in each frame image is the basis for obtaining its motion trajectory. Analyzing the characteristics of the basketball in the video image is the key to getting the basketball position. In the image, the color of the basketball has unique RGB component characteristics, and the size of the basketball and the grayscale difference between the circular edge and the background are the two most important features that distinguish it from the background information. The trajectory of the basketball obtained is shown in Figure 2.

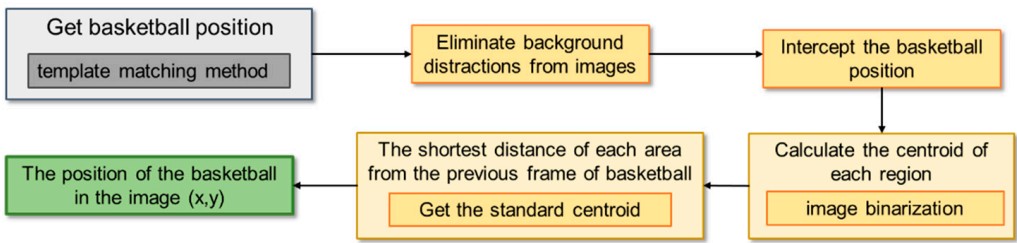

**Figure 2.** Basketball position acquisition process.

First, by analyzing the shape and color characteristics of the basketball, the first frame image is used as the image to be searched by the template matching method, and the basketball image is used as a template to match pixel by pixel on the image to be searched, and the basketball is obtained in the first frame image. Secondly, due to the background transformation and position change, it is not accurate to use the direct matching method to locate the basketball in the images after the second frame. Here, the interference of the background is eliminated by constructing the red channel. Thirdly, keep several limited areas including the basketball, calculate the centroid of the area, and the distance from the centroid to the basketball position in the first frame; the centroid with the smallest distance can be regarded as the position of the basketball. Finally, the video image is processed frame by frame to obtain a complete basketball trajectory.

*3.2. Determination of the Initial Position of the Basketball*

During the shooting process, the determination of the initial position of the basketball is the premise of accurately constructing the trajectory of the basketball. In this paper, image registration technology is used to determine the initial position of the basketball in the video. The image to be searched is a sub-image obtained by cutting the basketball in the first frame of the video, and the sub-image contains all the features of the target basketball. As shown in Figure 3, the first frame image in the shooting video is used for the target to

be searched. The task of the search algorithm is to find the position of the template image at the target to be searched, and use this position as the reference position of the basketball, so as to complete further operations on the remaining video.

Image to be searched

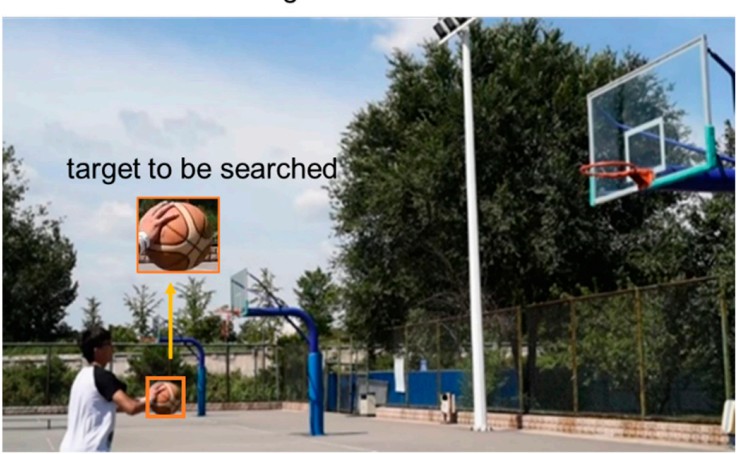

**Figure 3.** The searched image and the searched target.

Assuming that the size of the image to be searched is N × N, the size of the target to be searched is M × M, the image to be searched moves pixel by pixel on the image to be searched S, and the part where the target to be searched and the image overlap is called sub-image. S(i, j) are the coordinates of the pixel in the upper left corner of the sub-image in the image to be searched, which are called reference points, where the value ranges of i and j are both (1, N − M + 1). In order to improve the operation speed, this paper adopts the stacked sparse denoising auto-encoders (SSDA) to complete the matching degree detection [19]. At the same time, we use the formula to explain the principle of SSDA more intuitively. First, the absolute error value is defined as:

$$\varepsilon(i, j, m_k, n_k) = \left| S^{i,j}(m_k, n_k) - \hat{S}^{i,j}(i, j) - T(m_k, n_k) + \hat{T} \right| \tag{1}$$

$$\hat{S}(i, j) = \frac{1}{M^2} \sum_{m=1}^{M} \sum_{n=1}^{M} S^{i,j}(m, n) \tag{2}$$

$$\hat{T} = \frac{1}{M^2} \sum_{m=1}^{M} \sum_{n=1}^{M} T(m, n) \tag{3}$$

Take a constant threshold value $T_k$, randomly select pixel point $(m_k, n_k)$ in sub-picture $S^{i,j}(m, n)$, calculate the error value $\varepsilon(i, j, m_k, n_k)$ between it and the corresponding point in T, and then add up the difference value and the difference value of other point pairs. When the accumulated error exceeds $T_k$ for r times, the accumulation is stopped and the number r is recorded.

Define the detection surface of SSDA as shown in (4):

$$I(i, j) = r \tag{4}$$

The point $(i, j)$ corresponding to $I(i, j)$ with the largest value is used as the matching point $(x, y)$, so that the position point $(x, y)$ of the basketball in the first frame image is obtained as the initial position.

### 3.3. Real-Time Position Acquisition during Basketball Movement

When using the image registration technology to determine the initial position of the basketball, the image to be searched is cut out from the first frame of image, so the initial position is relatively accurate. However, since the position of the basketball and the color and shape features of the images of the second frame and later are constantly

changing, it is difficult to obtain the exact position of the target basketball when the same template is used to continue matching. First, keep the accurate position of the basketball in the first frame image and use it as the reference position. Then, by analyzing the color and shape characteristics of the basketball, using the large difference between the basketball color and the background, and using the RGB components to effectively remove the background interference, multiple areas containing the basketball are obtained. The position of the basketball can be determined through further operations, and the process is shown in Figure 4.

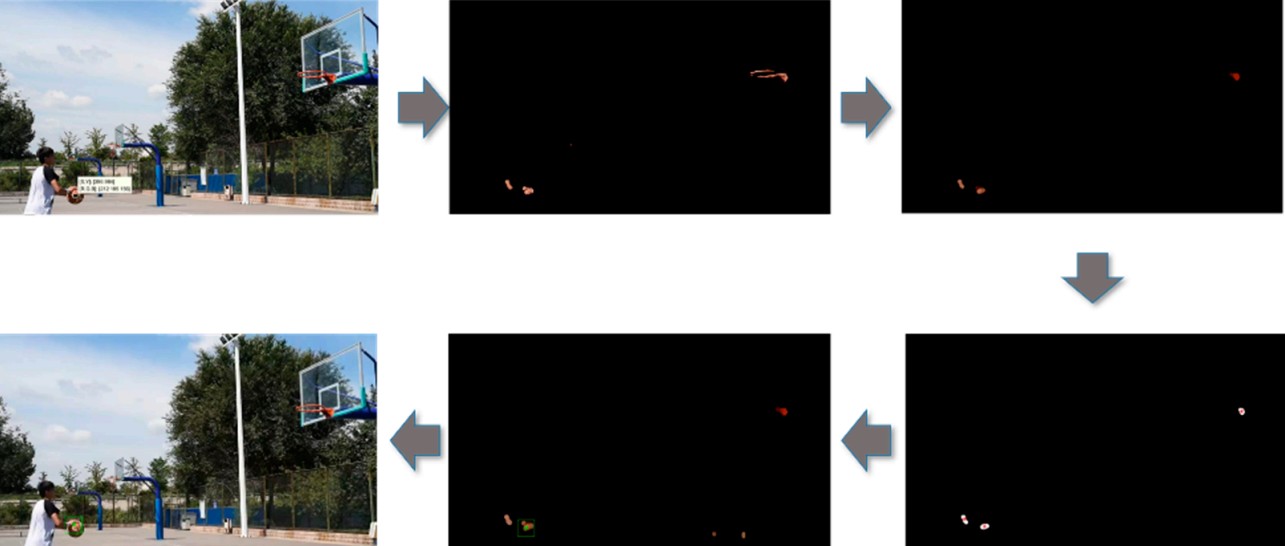

**Figure 4.** RGB component acquisition process of basketball position.

The RGB value refers to the brightness value that divides the three colors of red, green and blue into 256 levels, and the objects in each color image have different color characteristic RGB components. Since the red channel features of the basketball are the most obvious in the video images collected in this paper, by constructing the red channel, the interference of the sky, trees, ground, and testers can be eliminated, while most of the characteristics of the basketball and objects similar to the basketball can be retained. However, there are still some small interferences in the processed images, which are eliminated by the open operation [20]. For the image A and the target S, use A + S to express the use of S to dilate A, and use A − S to express the use of S to erode A, then the open operation relation of the image is as shown in (5) [21].

$$A \circ E = (A - E) + E \tag{5}$$

The image after background removal and opening operation is shown in Figure 5, which contains only four limited areas, of which the area 1 is the basketball, and the rest are interference areas.

Since in this video, the time interval between two adjacent frames is only 8.3 ms, the actual distance of the basketball in this time interval is actually very short, and the positional deviation of the basketball in the adjacent two frames is also small. We took the basketball position in the first frame image as the reference position for matching, then binarize the whole image and calculate the centroid of each region in Figure 5. After that, the distance of these centroids relative to the reference position is further calculated, and the centroid with the smallest distance is the coordinate position of the target basketball in this frame image, and the centroid calculation is shown in (6) [22].

$$M_{pq} = \int_{a1}^{a2} \int_{b1}^{b2} x^p y^q f(x, y) dx dy \tag{6}$$

The basketball position obtained by the centroid method is more accurate, and the data in the subsequent experimental results are all the basketball positions calculated by the centroid method. By integrating the basketball positions of each frame of images, the basketball trajectory of the whole shooting process is obtained. In addition, due to the possible deviation of the calculated basketball position, individual points may deviate from the original trajectory. At this time, if it is used as a sample for training, it will cause a large error. Therefore, for the trajectory data of the training samples, this paper uses smooth interpolation to interpolate the obtained trajectory sequence to obtain a smooth trajectory sequence that is closer to the real motion trajectory.

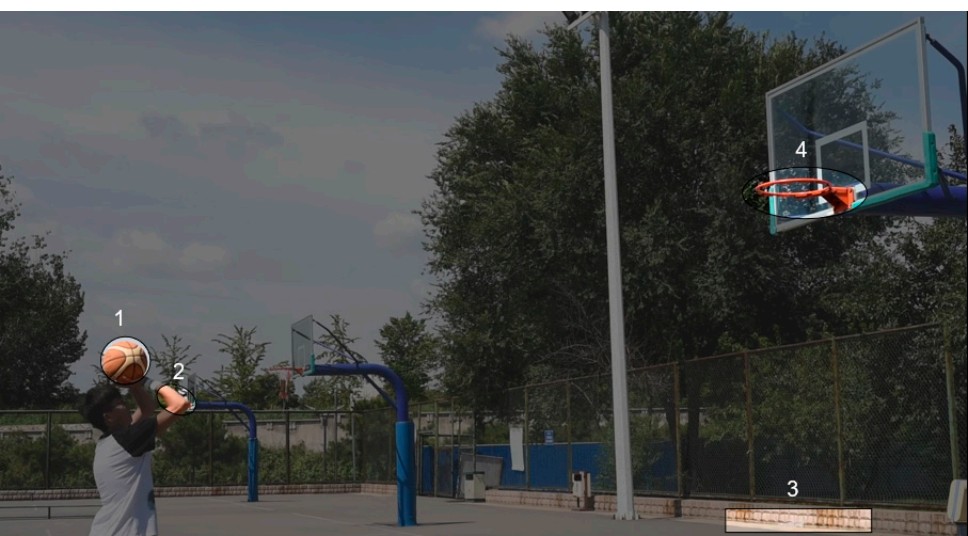

**Figure 5.** The shot image after constructing the red channel and opening operation.

### 3.4. Basketball Trajectory Feature Extraction

During the experiment, we divided the trajectory of the basketball into three stages, namely the starting stage, the shooting stage, opnm, and the landing stage after shooting. Among them, the starting stage refers to the force-generating stage before the basketball leaves the hand; the shooting stage refers to the movement period before the basketball leaves the hand and touches the basket. The landing stage refers to the landing period after the basketball touches the basket or reaches the same height as the basket. Since the position of the basket and the position of leaving the hand can be obtained through image processing, the data of the shooting stage can be obtained. Take the instantaneous position of the basketball leaving the palm as the starting point of the trajectory, and the position where the basketball reaches the backboard as the end point of the trajectory, and remove the motion data before the shot time and after reaching the backboard, as shown in Figure 6. After the trajectory data is obtained, feature data is constructed from the trajectory data, which specifically includes: trajectory pair data (x, y), distance $D = \sqrt{x^2 + y^2}$, time t to reach the fixed-point segment, basketball height y, etc. In the feature analysis, the shooting prediction results obtained by different feature combinations will be very different, so the prediction results of different feature combinations need to be further considered in the prediction.

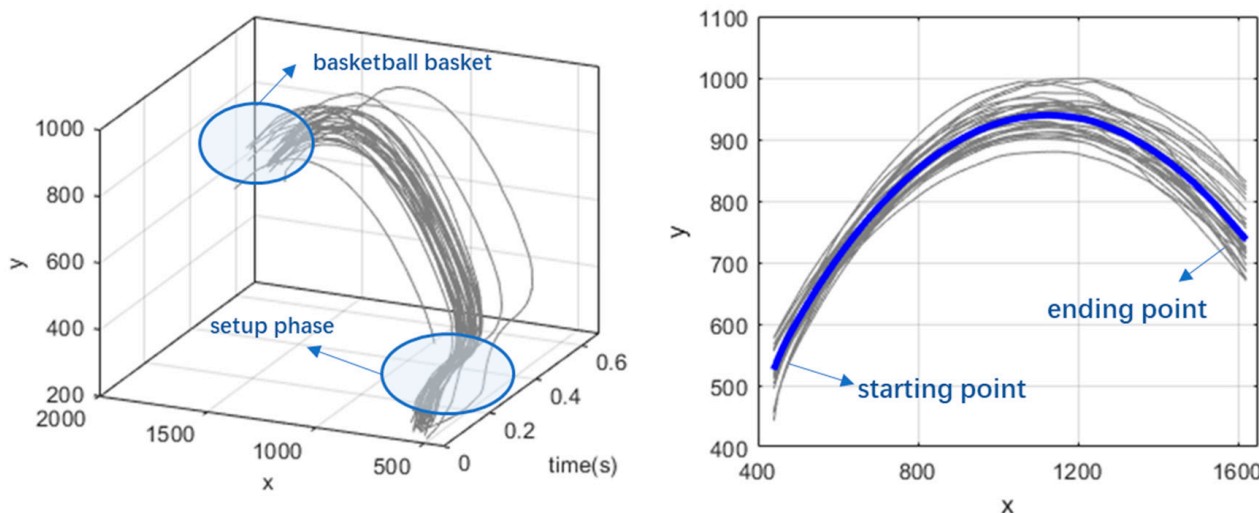

**Figure 6.** Basketball trajectory in shooting video.

## 4. Shooting Results Prediction

We conducted 55 shooting experiments, including 31 successful shots and 24 missed shots. To verify the superiority of our method, we divided the original dataset into two groups. The first set contains trajectory coordinate data for 16 successful and 12 missed shots. These data make up the training set to train the classifier. The second set contains trajectories coordinate data for 15 successful shots and 12 missed shots. These data constitute the test set to test the prediction of shooting results.

We divided the data into training set and test set after obtaining the basketball trajectory data. Then, we preprocessed the basketball trajectories in the training set. We interpolated the Y coordinate and time t separately. We first selected different feature combinations and then trained the predictive model through a machine learning classification algorithm. We fed the data from the test set into a prediction model to draw conclusions about whether to shoot or not. Finally, we compared the results with the actual results to obtain the prediction accuracy.

The execution of the classification algorithm determines the accuracy of the model predictions. To explore the best classification algorithm, this paper used four different machine learning algorithms to predict shooting results, including K-NearestNeighbor (KNN) [23–26], naive Bayes (NB) [23–25], linear discriminant analysis (LDA) [27,28], and classification and regression trees (CART) [29–31].

The purpose of shooting prediction is to predict the effect of shooting in advance based on the shooting information at the current moment. This can provide basic guidance for basketball players when they practice shooting. We need to achieve faster predictions by using as few trajectory data points as possible while maintaining the accuracy of shot predictions. Therefore, we need to balance the two key factors of accuracy and trajectory length. This paper discussed the prediction results under three feature combinations: time t and coordinates (x, y), time t and distance D, and time t and altitude Y.

### 4.1. Shot Prediction Based on Time and Motion Trajectory

Firstly, we used time t and basketball trajectory coordinates (x, y) as a combination of sample features to predict the shooting result, and the obtained prediction accuracy is shown in Figure 7a. We can see that the prediction accuracy of the four classification algorithms all show an upward trend when the number of trajectory points increases. Among the four classifiers, the CART decision tree algorithm exhibited the best stability and a higher accuracy. When the number of trajectory points reaches 20 (one-third of the entire trajectory), we can get the best prediction results. As the number of trajectory points increases, the accuracy remains around 85%. The prediction accuracy of the four classifiers

is higher than 70%, and the highest is 85%. Secondly, the results shown in Figure 7b are based on the feature combination [D, t], and the prediction accuracy after stabilization is 60%. Lastly, the feature combination used in the prediction result shown in Figure 7c is time t and height Y. When the number of trajectory points is greater than 12, we can see that the prediction accuracy of the CART decision tree algorithm is basically stable at about 85%, the prediction accuracy of the LDA algorithm is higher than 80%, and the prediction accuracy of the KNN and NB classifiers is higher than 70%. The overall prediction effect is similar to the prediction results based on time and trajectory features. According to the above three experiments, the CART algorithm has a good effect on the shot prediction. We conclude that the shooting prediction results based on time and trajectory features and time t or height Y features are better, and the overall accuracy can reach 85%. The prediction accuracy of experiment based on time and distance D is low, so it is not suitable for basketball shoot prediction. In addition, it can be seen that the accuracy of shot prediction based on time and trajectory features stabilizes after the 17th trajectory point, accounting for 39.53% of the entire basketball trajectory. The shot prediction based on time and distance D features can reach the steady state after the 12th trajectory point, accounting for 7.91% of the entire basketball trajectory. Therefore, the best feature combination scheme for shot prediction is the shot prediction based on time and height. At the same time, we can find that the method of this paper can be used to predict the shot result with high accuracy by obtaining the data of the previous period during the shooting process of the athlete.

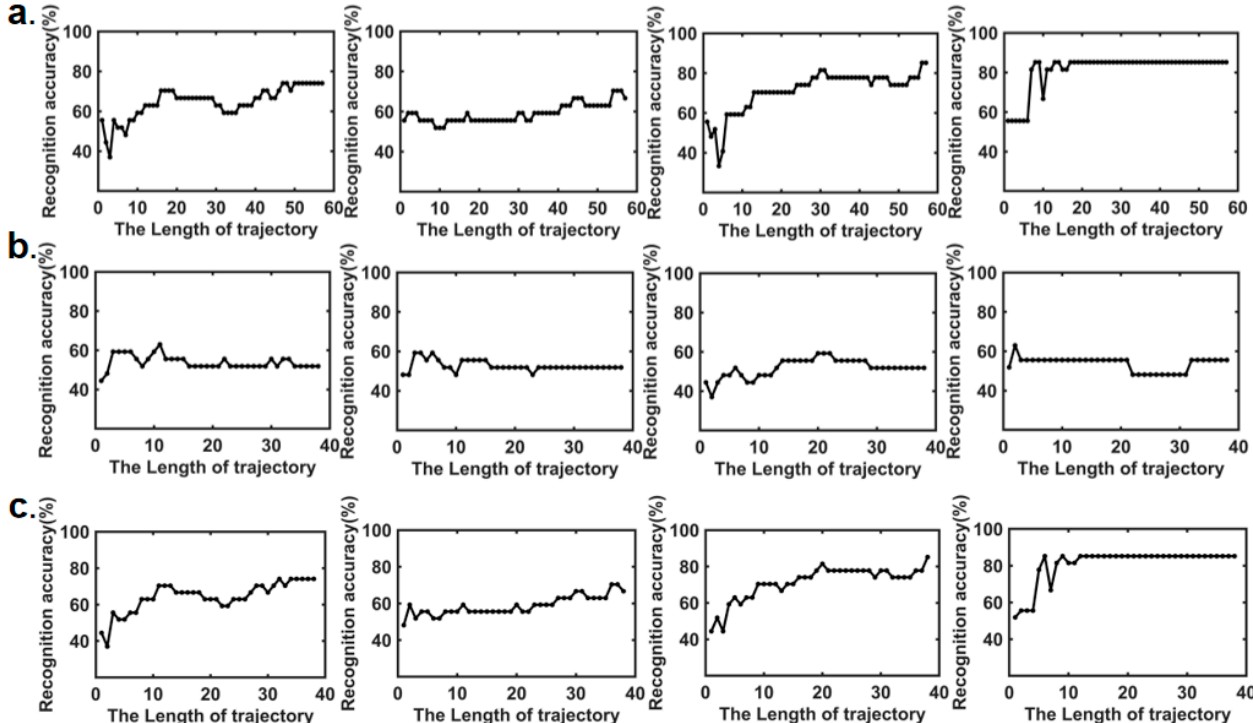

**Figure 7.** Prediction accuracy based on (**a**) time t and coordinates (x, y), (**b**) time t and distance D, and (**c**) time and height Y of different trajectory points with KNN, NB, LDA, and CART.

*4.2. The Effect of the Density of Smooth Interpolation on the Prediction Results*

We can find that the number of basketball trajectory points obtained after image processing depends on the density of trajectory interpolation. The more the number of trajectory points, the smoother the trajectory curve that is obtained. The trajectory curve thus obtained is closer to the real basketball trajectory. Therefore, to explore this effect, in addition to the basic 43-point trajectory length mentioned above, we also introduced other interpolation points for prediction. Finally, we obtained the prediction results for

28, 79, and 116 points after interpolation of the total trajectory after stabilization. The final prediction results are shown in Figure 8.

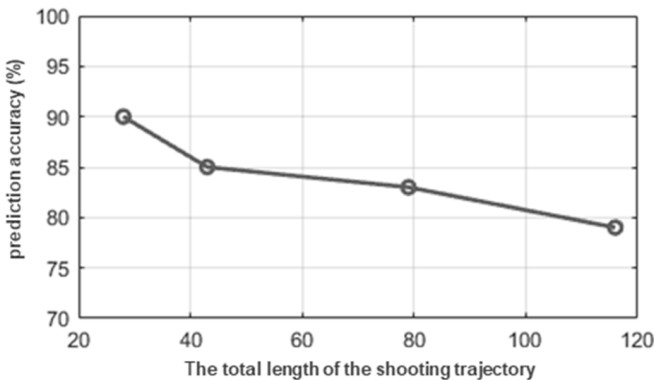

**Figure 8.** Prediction results under different interpolation points.

It can be seen that when the interpolation density is high (the total number of points is 116), the prediction accuracy of the CART algorithm is improved, and the prediction accuracy increases to 90%. After reducing the interpolation density (the total number of points is 43), the prediction accuracy of the algorithm is significantly reduced, and the prediction accuracy is 85%. When the interpolation density reaches 28 total trajectory points, the prediction accuracy is reduced to 79%. Therefore, it can be concluded that the accuracy of shot prediction is positively correlated with the interpolation density, which also reflects that the prediction accuracy may be related to the smoothness of the trajectory and the video frame rate.

## 5. Discussion

Through the experimental results, we can find that the best combination of prediction features is the prediction method based on the height and time characteristics of basketball movement. In real prediction modeling, the optimal prediction data length is about one-third of the overall trajectory motion process. This time node can basically ensure that the overall prediction accuracy is stable and has a faster prediction time. Our proposed method can not only capture the free-throw trajectory of the basketball accurately, but can also predict the whole shooting trajectory based on partly trajectory. The acceleration of the basketball during the shooting process was collected by an accelerometer. The experimental results showed a shooting accuracy of 86.3%. However, the method extracts insufficient feature information, which requires a lot of data to improve the accuracy. The use of wearable sensors for data collection and analysis helps improve shooting accuracy by providing feedback on parameters such as bounce speed and release angle. However, the shooting accuracy can only reach up to 71%. In addition, the basketball shooting gesture recognition method based on image processing and machine learning collects basketball players' action pose data and extracts multidimensional motion pose features from the time and frequency domains. The average shot hit rate of this method can reach 91.2%.

The research in this paper also has certain limitations. The object of our study is the free-throw shooting, which limits the player standing at the free-throw line. In free-throw shooting, we specify the basketball player to shoot directly towards the basket at the free-throw line to avoid the ball's displacement in the third dimension. When the player changes position to shoot again, our method no longer applies. At the same time, since the image-based trajectory extraction method is easily affected by environmental factors, its accuracy has further potential for improvement. Therefore, our next step is to improve our algorithm to reduce the influence of environmental factors on our results.

## 6. Conclusions

This paper overcomes the problem of capturing the position of a basketball based on a shooting video and delineating its motion trajectory. A method is proposed to obtain the trajectory of the basketball by analyzing the color and shape features of the basketball and registering them with the video. Based on the coordinates of the movement trajectory of a large number of basketballs, the prediction model of whether a shot was made or not is trained. For different shooting video backgrounds, this method can effectively obtain the trajectory of a basketball and has good universality.

The basketball position obtained by the basketball trajectory calculation method proposed in this paper is very close to the actual basketball position. The trained shooting prediction model has high accuracy when the number of training samples is large enough. Compared with other methods, the method proposed in this paper is more simple and more effective, and the trained prediction model can be successfully applied in shooting prediction, and it also has certain novelty.

In future work, the algorithm will be further optimized to study how to use fewer coordinates to reduce the error of basketball positioning and prediction results, so that it can capture the basketball's position in a more complex background and obtain more accurate results.

**Author Contributions:** Conceptualization, Y.Z., C.L. and G.Z.; methodology, X.Z., M.Y. and J.L.; software, X.Z. and Q.Z.; validation, Y.Z. and C.B.; formal analysis, X.Z., M.Y. and Q.Z.; data curation, J.L. and C.L.; writing—original draft preparation, X.Z. and M.Y.; writing—review and editing, Y.Z. and J.L.; visualization, X.Z. and Z.W.; supervision, Y.Z., C.L. and G.Z.; project administration, Y.Z.; funding acquisition, Y.Z. and G.Z. All authors have read and agreed to the published version of the manuscript.

**Funding:** This work was supported by the National Natural Science Foundation of China (Grant No. 61873307), the Hebei Natural Science Foundation (Grant Nos. F2020501040, F2021203070, F2022501031), the Fundamental Research Funds for the Central Universities under Grant No. 2123004, and the Administration of Central Funds Guiding the Local Science and Technology Development (Grant No. 206Z1702G), the Shenzhen Science and Technology Innovation Commission (SZSTI) (Grant No. JCYJ20190808181803703).

**Institutional Review Board Statement:** Not applicable.

**Informed Consent Statement:** Not applicable.

**Data Availability Statement:** The data presented in this study are publicly available by visiting: https://github.com/dlj0214/shooting-prediction (accessed on 3 September 2022).

**Conflicts of Interest:** The authors declare no conflict of interest.

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
