# Peer review of "Shooting Prediction Based on Vision Sensors and Trajectory Learning"

_applsci, doi:10.3390/app121910115_

Round 1
Reviewer 1 Report
Authors tried to use vision for prediction of success in basketball shooting. Movement of ball is in three dimensional area, My concern is that, How two dimensional vision can give us 90% accuracy when we do not have information of third dimesion? Other concern is that, how prediction of trajectory after shooting can help the player to improve his/her prformance?
I propose that aim of research needs modification. Better prediction needs at least view from two different points.
Reviewer 2 Report
This paper is interesting, however, some issue should be corrected
a. In the introduction part, authors should give some simple discussions to show the advantages and disadvantages of the existing results.
b. Please add organized of paper in the end of introduction
c. The authors have to further improve the technical writing and presentation of the paper.
d. Please add refs for equation
e. Fig 7, Fig 8, Fig 9 are low quality, please not printscreen, author can drawn using software. I suggest author drawn with one graphic for comparative
f. please give more discussion for section 5. Compare with simlliar studies, limitation of this study etc
g. Throughout the manuscript grammatical errors and typos should be thoroughly checked
Round 2
Reviewer 1 Report
Dear Respected Authors
Figure 5 needs modification. When we have two dimensional analysis, it is better that position of the camera be accurately in center of distance between player and basket. But in figure 5, it seems more close to player.
Figure 6 has two subfigures. It will be better that they labeled separately.
In left figure, Authors can show time via other method instead of the third axis.
It worth to be mentioned that in the paper will be emphasized that research is two-dimensional analysis.